# SELECTIVE SAMPLING FOR ACCELERATING TRAINING OF DEEP NEURAL NETWORKS

## ABSTRACT

We present a selective sampling method designed to accelerate the training of deep neural networks. To this end, we introduce a novel measurement, the *minimal margin score* (MMS), which measures the minimal amount of displacement an input should take until its predicted classification is switched. For multi-class linear classification, the MMS measure is a natural generalization of the margin-based selection criterion, which was thoroughly studied in the binary classification setting. In addition, the MMS measure provides an interesting insight into the progress of the training process and can be useful for designing and monitoring new training regimes. Empirically we demonstrate a substantial acceleration when training commonly used deep neural network architectures for popular image classification tasks. The efficiency of our method is compared against the standard training procedures, and against commonly used selective sampling alternatives: Hard negative mining selection, and Entropy-based selection. Finally, we demonstrate an additional speedup when we adopt a more aggressive learning-drop regime while using the MMS selective sampling method.

## 1 INTRODUCTION

Over the last decade, deep neural networks have become the machine learning method of choice in a variety of application domains, demonstrating outstanding, often close to human-level, performances in a variety of tasks. Much of this tremendous success should be attributed to the availability of resources; a massive amount of data and compute power, which in turn fueled the impressive and innovative algorithmic and modeling development. However, resources, although available, come with a price. Data in the big data era is available, but reliable labeled data is always a challenge, and so are the ETL (Extract-Transform-Load) processes, data transfer, and storage. With the introduction of GPUs, compute power is readily available, making the training of deep architectures feasible. However, the training phase, which to a large extent, relies on stochastic gradient descent methods, requires a large number of computational resources as well as a substantial amount of time. A closer look at the compute processes highlights the fact that there is a significant difference in the compute effort between the inference (forward pass) and the model update (back-propagation) where the latter being far more demanding. The implication is evidenced by the performance charts that hardware manufactures publish, where performance matrices such as throughput (e.g. image per second) are up to 10x better at inference vs. training for popular deep neural network architectures.

In this paper, we address the computing challenge. Specifically, we suggest a method to select for the back-propagation pass only those instances that accelerate the training convergence of the deep neural network, thus speeding up the entire training process. The selection process is continuously performed throughout the training process at each step and in every training epoch. Our selection criterion is based on computations that are calculated anyhow as an integral part of the forward pass, thus taking advantage of the "cheaper" inference compute.

## 2 PREVIOUS APPROACHES

Accelerating the training process is a long-standing challenge that was already addressed by quite a few authors. A common approach is to increase the batch size, thus mitigating the inherent time load. This approach represents a delicate balance between available compute ingredients (e.g. memory

size, bandwidth, and compute elements). Interestingly, increasing the batch size not only impacts the computational burden but may also impact the final accuracy of the model (Goyal et al., 2017; Jia et al., 2018; Ying et al., 2018).

Sample selection is another approach that has been suggested to accelerate the training. The most notable one is probably the hard negative mining (Schroff et al., 2015) where samples are selected by their loss values. The underlying assumption is that samples with higher losses have a significant impact on the model. Most of the previous work that utilized this approach was mainly aimed at increasing the model accuracy, but the same approach can also be used to accelerate training. Recent works employ selection schemes that examine the importance of the samples (Alain et al., 2015; Loshchilov & Hutter, 2015). During the training, the samples are selected based on their gradient norm, which in turn leads to a variance reduction in the stochastic gradients. Inspired by the batch size approach, a recent work by Katharopoulos and Fleuret (Katharopoulos & Fleuret, 2018) uses selective sampling to choose the training samples that reduce the gradient variance, rather than increasing the size of the batch.

Our work is inspired by the *active learning* paradigm that utilizes selective sampling to choose the most useful examples for training. In active learning, the goal is to reduce the cost of labeling the training data by querying the labels of only carefully selected examples. Thus, unlike the common supervised learning setting, where training data is randomly selected, in active learning, the learner is given the power to ask questions, e.g. to select the most valuable examples to query for their labels. Measuring the training value of examples is a subject of intensive research, and quite a few selection criteria have been proposed. The approach most related to our work is the *uncertainty sampling* (**?**)lewis1994sequential), where samples are selected based on the uncertainty of their predict labels. Two heavily used approaches to measure uncertainty are entropy-based and margin-based (Settles, 2009). In the entropy-based approach (Lewis & Catlett, 1994), uncertainty is measured by the entropy of the posterior probability distribution of the labels, given the sample. Thus, a higher entropy represents higher uncertainty with respect to the class label. This approach naturally handles both binary and multi-class classification settings, but it relies on an accurate estimate of the (predicted) posterior probabilities. In the margin-based approach(Tong & Koller, 2001; Campbell et al., 2000), uncertainty is measured by the distance of the samples from the decision boundary. For linear classifiers, several works (Dasgupta, 2006; Balcan et al., 2007) gave theoretical bounds for the exponential improvement in computational complexity by selecting as few labels as possible. The idea is to label samples that reduce the *version space* (a set of classifiers consistent with the samples labeled so far) to the point where it has a diameter at most $\varepsilon$ (c.f (Dasgupta, 2011)). This approach was proven to be useful also in non-realizable cases (Balcan et al., 2007). However, generalizing it to the multi-class setting is less obvious. Another challenge in adapting this approach for deep learning is how to measure the distance to the intractable decision boundary. Ducoffe and Precioso (Ducoffe & Precioso, 2018) approximate the distance to the decision boundary using the distance to the nearest adversarial examples. The adversarial examples are generated using a Deep-Fool algorithm (Moosavi-Dezfooli et al., 2016). The suggested DeepFool Active Learning method (DFAL) labels both, the unlabeled samples and the adversarial counterparts, with the same label.

Our selection method is also utilizing uncertainty sampling, where the selection criterion is the closeness to the decision boundary. We do, however, consider the decision boundaries at the (last) fully-connected layer, i.e. a multi-class linear classification setting. To this aim, we introduce the *minimal margin score* (MMS), which measures the distance to the decision boundary of the two most competing predicted labels. This MMS serves us as a measure to score the assigned examples. A similar measure was suggested by Jiang et al. (Jiang et al.) as a loss function and a measure to predict the generalization gap of the network. Jiang et al. used their measure in a supervised learning setting and applied it to all layers. In contrast, we apply this measure only at the last layer, taking advantage of the linearity of the decision boundaries. Moreover, we use it for selective sampling, based solely on the assigned scores, namely without the knowledge of the true labels. The MMS measure can also be viewed as an approximation measure for the amount of perturbation needed to cross the decision boundary. Unlike the DFAL algorithm, we are not generating additional (adversarial) examples to approximate this distance but rather calculate it based on the scores of the last-layer.

Although our selective sampling method is founded by active learning principles, the objective is different. Rather than reducing the cost of labelling, our goal is to accelerate the training. Therefore,

we are more aggressive in the selection of the examples to form a batch group at each learning step, at the cost of selecting many examples at the course of training.

The rest of the paper is organized as follows. In section 3, we present the MMS measure and describe our selective sampling algorithm and discuss its properties. In Section 4 we present the performances of our algorithm on the common datasets CIFAR10 and CIFAR100 (Krizhevsky et al., 2009) and compare results against the original training algorithm and hard-negative sampling. We demonstrate additional speedup when we adopt a more aggressive learning-drop regime. We conclude at Section 5 with a discussion and suggestions for further research.

## 3   ACCELERATING TRAINING USING MINIMAL MARGIN SCORE SELECTION

As mentioned above, our method is based on the evaluation of the minimal amount of displacement a training sample should undergo until its predicted classification is switched. We call this measure *minimal margin score* (MMS). This measure depends on the best and the 2nd best scores achieved for a sample. Our measure was inspired by the margin-based quantity suggested by Jiang et al. (Jiang et al.) for predicting the generalization gap of a given network. However, in our scheme, we apply our measure to the output layer, and we calculate it linearly with respect to the input of the last layer. Additionally, unlike (Jiang et al.), we do not care about the true label, and our measure is calculated based on the best and the 2nd best NN scores.

An illustrative example, demonstrating the proposed approach, is given in Figure 1. In this example, a multi-class classification problem is composed of three classes: Green, Red, and Blue along with three linear projections: $\mathbf{w}_1, \mathbf{w}_2$, and $\mathbf{w}_3$, respectively. The query point is marked by an empty black circle. The highest scores of the query point are $s^1$ and $s^2$ (assuming all biases are 0's), where $s^1 > s^2$ and $s^3$ is negative (not marked). Since the best two scores are for the Green and Red classes, the distance of the query point to the decision boundary between these two classes is $d$. The magnitude of $d$ is the MMS of this query point.

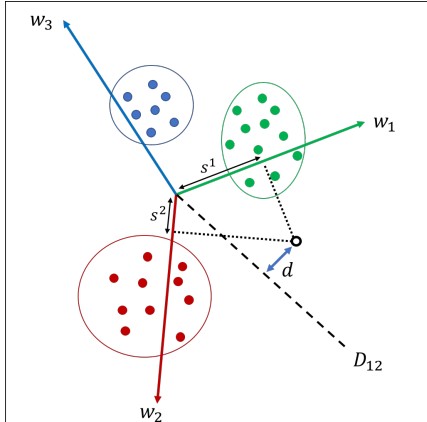

Figure 1: Illustrative example of the MMS measure. For more details see text.

Formally, let $\mathcal{X} = \{\mathbf{x}_1, ..., \mathbf{x}_B\}$ be a large set of samples and $\mathbf{y}_i = F(\mathbf{x}_i; \theta) \in \mathcal{Y}$ be input to the last layer of the neural network $F$. Assume we have a classification problem with $n$ classes. At the last layer the classifier $f$ consists of $n$ linear functions: $f_i : \mathcal{Y} \to \mathbb{R}$ for $i = 1 \ldots n$ where $f_i$ is a linear mapping $f_i = \mathbf{w}_i^T \mathbf{y} + b_i$. For sample $\mathbf{x}_k \in \mathcal{X}$, the classifier predicts its class label by the maximal score achieved: $\ell_k = \arg\max_i f_i(F(\mathbf{x}_k; \theta)) = \arg\max_i f_i(\mathbf{y}_k)$. Denote the sorted scores of $\{f_i(\mathbf{y}_k)\}_{i=1}^n$ by $(s_k^{i_1}, s_k^{i_2}, \cdots, s_k^{i_n})$ where $s_k^{i_j} \geq s_k^{i_{j+1}}$ and $s_k^{i_j} = f_{i_j}(\mathbf{y}_k)$. The classifier $f_{i_1}(\mathbf{y}_k)$ gave the highest score and $f_{i_2}(\mathbf{x}_k)$ gave the second highest score. The *decision boundary* between class $i_1$ and class $i_2$ is defined as:

$$D_{12} = \{\mathbf{y} | \ f_{i_1}(\mathbf{y}) = f_{i_2}(\mathbf{y})\}$$

Using this definition, the confidence of the the predicted label $i_1$ of point $\mathbf{x}_k$ is determined by the distance of $\mathbf{y}_k$ to the decision boundary $D_{12}$, namely the minimal distance of $\mathbf{y}_k$ to $D_{12}$:

$$d_k = min_{\delta \mathbf{y}} ||\delta \mathbf{y}|| \quad \text{s.t.} \ (\mathbf{y}_k + \delta \mathbf{y}) \in D_{12}$$

It is easy to show (see Appendix A) that

$$d_k = \frac{s_k^{i_1} - s_k^{i_2}}{\|\mathbf{w}_{i_1} - \mathbf{w}_{i_2}\|}$$

The distance $d_k$ is the *Minimal Margin Score* (MMS) of point $\mathbf{x}_k$. The larger the $d_k$, the more confident we are about the predicted label. Conversely, the smaller the $d_k$, the less confident we are about the predicted label $i_1$. Therefore, $d_k$ can serve as a confidence measure for the predicted labels. Accordingly, the best points to select for the back-propagation step are the points whose MMS are the smallest.

Our implementation consists of a generic but yet simple online selective sampling method as a preliminary part of the training optimization step. Specifically, at each training step, a selective batch of size $b$ is produced out of a $10x$ larger batch before using it in the SGD optimization routine. At a specific iteration, we first apply a forward pass on a batch of size $B$, producing the prediction scores. Secondly, we select $b$ samples ($b \ll B$) whose MMS measures are the smallest. This procedure is summarized in Algorithm 1.

---

**Algorithm 1:** Selection by Minimal Margin Scores

---

**Require:** Inputs $\mathcal{X} = \{\mathbf{x}_i\}_{i=1}^{B}$ , $F(\cdot; \theta_0)$ - Training model, b - batch size

  $t \leftarrow 1$

  **repeat**

    $\mathcal{Y} \leftarrow F(\mathcal{X}; \theta_{t-1})$             forward pass on batch of size B

    $MMS \leftarrow d(\mathcal{Y})$             calculates the Minimal Margin Scores of $\mathcal{Y}$

    $S \leftarrow \text{sort\_index}(MMS, b)$      stores the index of the $b$ smallest scores

    $\mathcal{X}_b = \{\mathbf{x}_i | \ i \in \mathcal{S}\}$          subset of $\mathcal{X}$ of of size b

    $\theta_t \leftarrow sgd\_step(F(\mathcal{X}_b; \theta_{t-1}))$    back prop. with batch of size b

    $t \leftarrow t + 1$

  **until** reached final model accuracy

---

## 4 EXPERIMENTS

In this section[1], we empirically examined the performance of the proposed selection scheme. As a baseline, we compared our results against the original training algorithm using uniform sampling from the dataset. Additionally, we compared the MMS method against two popular schemes: hard-negative mining which prefers samples with low prediction scores, and entropy-based uncertainty sampling. Our experimental workbench was composed of the common visual datasets CIFAR10 and CIFAR100 (Krizhevsky et al., 2009) that consist of $32 \times 32$ color images in 10 or 100 classes and has 50,000 training examples and 10,000 test examples.

**Hard negative samples.** For this experiment, we implemented the "NM-samples" procedure (Hoffer et al.), similar to the classical methods for "hard-negative-mining" used by machine-learning practitioners over the years (Yu et al., 2018; Schroff et al., 2015). We used the cross-entropy loss as a proxy for the selection, in which the highest loss samples were selected. We denote this approach as HNM-samples.

On both datasets we used ResNet-44 (He et al., 2016) and WRN-28-10 (Zagoruyko & Komodakis, 2016) architectures, respectively. To compare our MMS selection scheme against the baseline and HNM, we applied the original hyper-parameters and training regime using batch-size of 64. In addition, we used the original augmentation policy as described in He et al. (2016) for ResNet-44, while adding cutout (DeVries & Taylor, 2017) and auto-augment (Cubuk et al., 2018) for WRN-28-10. Optimization was performed for 200 epochs (equivalent to $156K$ iterations) after which baseline accuracy was obtained with no apparent improvement.

---

[1]All experiments were conducted using PyTorch framework, and the code is publicly available at `https://github.com/paper-submissions/mms-select`.

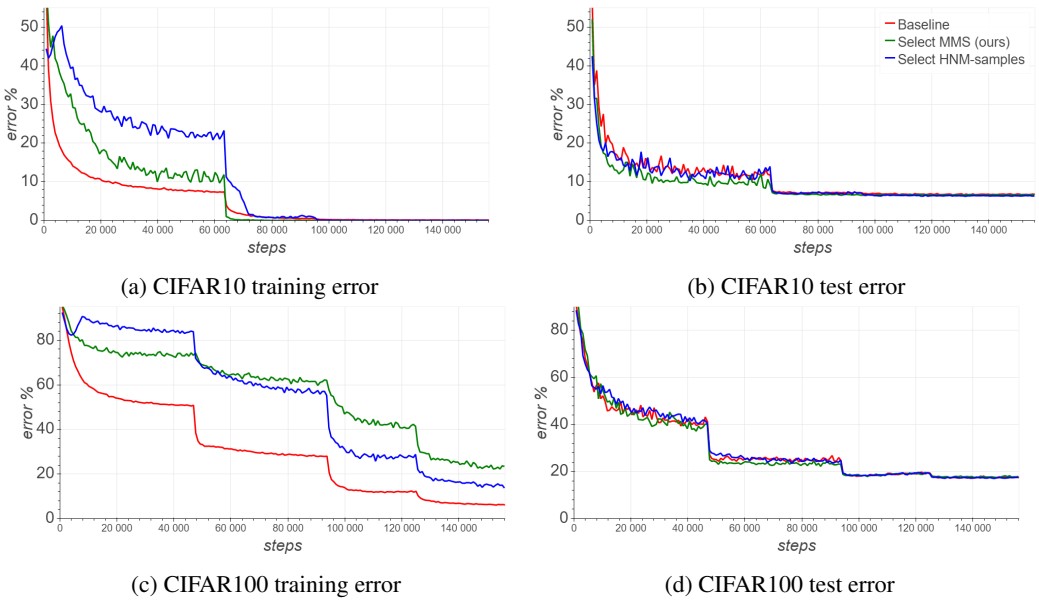

Figure 2: Training and test error (ResNet44, CIFAR10 and CIFAR100, WRN-28-10). Comparing vanilla training, HNM-samples selection (hard negative sampling), and MMS (our) selection

**CIFAR10.** For the CIFAR10 dataset, sampling with the MMS scheme obtained significantly lower error compared to the baseline and the HNM-samples throughout the entire training progress ($>$ $25\% - 30\%$ on average). The test results are depicted in Figure 2b. Furthermore, the use of MMS provides a slight improvement of 0.1% in the final test accuracy as well as a clear indication of a faster generalization compared to the baseline and the HNM schemes.

**CIFAR100.** Inspired by the results on CIFAR10 using the MMS method, we continued to evaluate performance on a more challenging 100 classes dataset. The MMS method obtained a non-negligible error decrease, particularly after the first learning-rate drop ($> 5\% - 10\%$ on average) as can be seen in Figure 2d. On the other hand, we did not observe similar behaviour using the HNM and the baseline schemes, similarly as in CIFAR10.

### 4.1 MEAN MMS AND TRAINING ERROR

To estimate the MMS values of the selected samples during training, we defined the mean MMS in a training step as the average MMS of the first 10 selected samples for the batch. This was compared to the mean MMS of the samples selected by the baseline and the HNM methods.

Figure 3 present the trace of the mean MMS that was recorded at the experiments presented in Figure 2 in the course of training. The mean MMS of the suggested scheme remains lower compared to the baseline in most of the training process. We argue that this behaviour stems from the nature of the uncertain classification with respect to the selected samples. This result suggests that there are "better" samples to train the model on rather than selecting the batch randomly. Interestingly, the HNM method obtained a similar mean MMS at the early stages of training, On the other hand, the HNM method resulted in a similar mean MMS as our suggested method during the training, but it increases after the learning-rate drop, and it deviates from the MMS scores obtained by our method. Lower mean MMS scores resemble a better (more informative) selected batch of samples. Hence, we may conclude that the batches selected by our method, provides a higher value for the training procedure vs. the HNM samples. Moreover, the mean MMS trace is monotonically increasing as the training progress, and it flattens when training converges.

All the selective sampling methods that we tested (HNM, entropy-based, and our MMS method), yielded a significantly higher error throughout the training (Figures 2a, 2c, 4). This coincides with

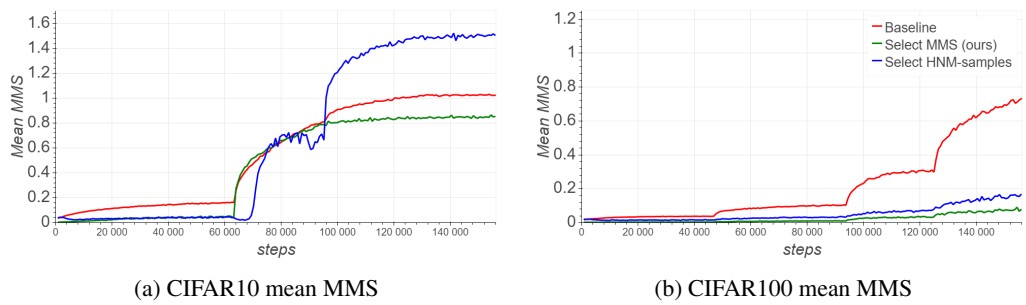

(a) CIFAR10 mean MMS  (b) CIFAR100 mean MMS

Figure 3: mean MMS of the samples selected by three methods: baseline, HNM-samples, and MMS.

the main theme of selective sampling that strive to focus training on the more informative points. However, training loss can be a poor proxy to this notion. For example, the selection criterion of the HNM favours high loss scores, which obviously increases the training error, while our MMS approach select uncertain points, some of which might be correctly classified, others might be miss-classified by a small margin (low absolute loss scores), but they are all close to the decision boundary, and hence useful for training. Evidently, the mean MMS provides a clearer perspective into the progress of training and usefulness of the selected samples.

## 4.2 SELECTIVE SAMPLING USING ENTROPY MEASURE

Additionally, we tested the entropy-based selective sampling, which is a popular form of uncertainty sampling. We select the examples with the largest entropy, thus the examples with the most class overlap, forming a training batch of size 64 out of a $10x$ larger batch. We compared performances with the vanilla training and the MMS selection method, using the same experimental setting.

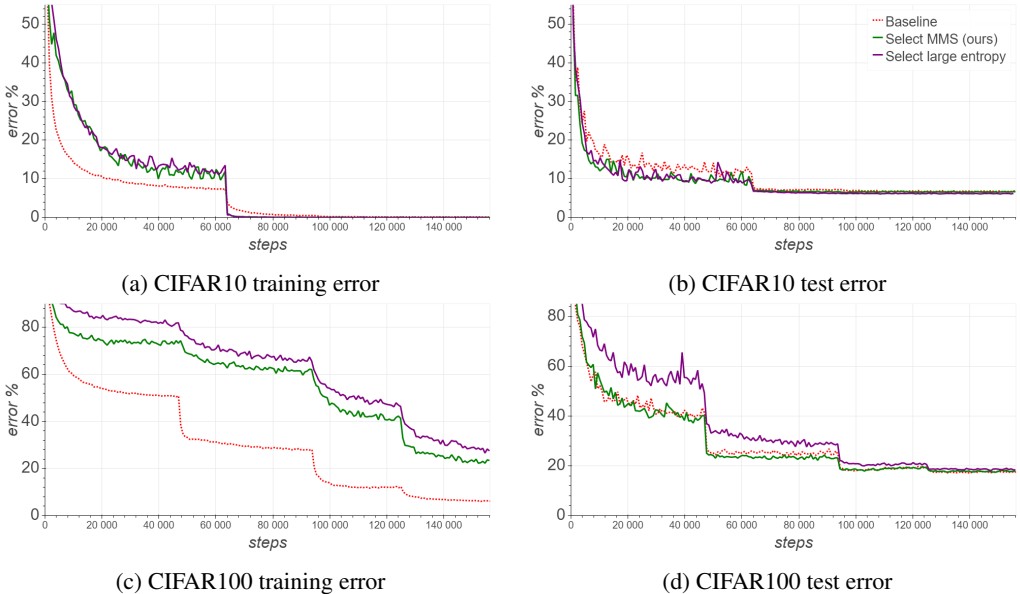

(a) CIFAR10 training error  (b) CIFAR10 test error

(c) CIFAR100 training error  (d) CIFAR100 test error

Figure 4: vanilla vs. entropy-based selection: Training and test error (ResNet44, CIFAR10 and WRN-28-10, CIFAR100).

This experiment shows (see Figure 4) that for a small problem as CIFAR10, this selection method is efficient as our MMS. However, as CIFAR100, inducing a more challenging task, this method fails. This entropy measure relies on the uncertainty of the posterior distribution with respect to the examples class. We consider this as an inferior method for selection. Also, as the ratio between the batch size and the number of classes increases, this measure becomes less accurate. Finally, as the number of classes grows, as in CIFAR100 compared to CIFAR10, the prediction scores signal has a

longer tail with less information, which also diminishes its value. The last assumption is not valid for our method as we measure based on the two best predictions.

### 4.3 ADDITIONAL SPEEDUP VIA AN AGGRESSIVE LEANING-RATE DROP REGIME

The experimental results have led us to conjecture that we may further accelerate training using the MMS selection, by applying an early learning-rate drop. To this end, we designed a new, more aggressive leaning-rate drop regime. Figure 5 present an empirical evidence to our conjecture that with the MMS selection method we can speed up training while preserving final model accuracy.

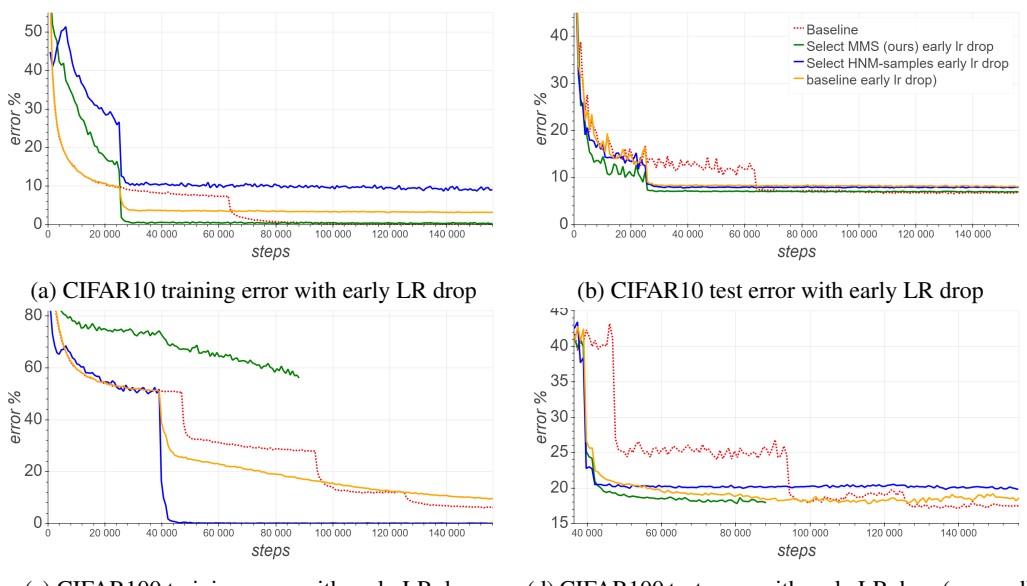

(a) CIFAR10 training error with early LR drop

(b) CIFAR10 test error with early LR drop

(c) CIFAR100 training error with early LR drop

(d) CIFAR100 test error with early LR drop (zoomed)

Figure 5: Training and test accuracy (ResNet44, CIFAR10 and WRN-28-10, CIFAR100). Comparing vanilla training, HNM-samples selection (hard negative mining), and MMS (our) selection method using a faster regime. We plot the regular regime baseline (dotted) for perspective. **The MMS selection method achieves final test accuracy at a reduces number of training steps**.

**CIFAR10.** For CIFAR10 and ResNet-44 we used the original learning rates $\eta = \{0.1, 0.01, 0.001, 0.0001\}$ while decreasing them at steps $\{24992, 27335, 29678\}$ equivalent to epochs $\{32, 35, 38\}$ with batch of size 64. As depicted in Figure 5b, we can see that indeed our selection yields validation accuracy that is similar to the one obtained using the original training regime, in a much earlier step. As described in table 1, training with our selection scheme almost reached final model accuracy in considerably less training steps as originally suggested. Specifically, we reached $93\%$ accuracy after merely $44K$ steps (a minor drop of $0.25\%$ compared to the baseline). We also apply the early drop regime to the baseline configuration as well as the HNM-samples. Both failed to reach the desired model accuracy while suffering from degradation of $1.57\%$ and $1.22\%$ for the baseline and HNM-samples, respectively.

Table 1: Test accuracy (Top-1) results for CIFAR10/100. We compare model accuracy using our training scheme and early learning-rate drop as described in section 4.3. We emphasize the reduces number of steps required reaching this accuracy using our MMS method.

| Network | Dataset | Steps | | Accuracy | |
|---|---|---|---|---|---|
| | | Baseline | Ours | Baseline | Ours |
| ResNet-44 | CIFAR10 | 156K | **44K** | 93.24% | 93% |
| WRN-28-10 | CIFAR100 | 156K | **80K** | 82.26% | 82.2% |

**CIFAR100.** Similarly, we applied the early learning-rate drop scheme for CIFAR100 and WRN-28-10, using $\eta = \{0.1, 0.02, 0.004, 0.0008\}$ and decreasing steps $\{39050, 41393, 43736\}$ equivalent to epochs $\{50, 53, 56\}$ and batch of size 64. As depicted in Figure 5d, accuracy reached $82.2\%$ with a drop of $0.07\%$ compared to baseline with almost halving the baseline required steps (from $156K$ to 80K steps). On the other hand, the baseline and the HNM-samples configurations failed to reach the desired accuracy after applying a similar early drop regime similarly to CIFAR10. The degradation for the HNM-samples approach was the most significant, with a drop of $2.97\%$ compared to the final model accuracy, while the baseline drop was approximately of $1\%$.

## 5 DISCUSSION

We presented a selective sampling method designed to accelerate the training of deep neural networks. Specifically, we utilized uncertainty sampling, where the criterion for selection is the distance to the decision boundary. To this end, we introduced a novel measurement, the *minimal margin score* (MMS), which measure the minimal amount of displacement an input should take until its predicted classification is switched. For multi-class linear classification, the MMS measure is a natural generalization of the margin-based selection criterion, which was thoroughly studied in the binary classification setting. We demonstrate a substantial acceleration for training commonly used DNN architectures for popular image classification tasks. The efficiency of our method is compared against the standard training procedures, and against commonly used selective sampling methods: Hard negative mining selection, and Entropy-based selection. Furthermore, we demonstrate an additional speedup when we adopt a more aggressive learning-drop regime.

Tracking the MMS measure throughout the training process provides an interesting insight into the training process. Figure 3 demonstrates that the MMS measure is monotonically increasing, even when training and validation errors are flattening. Subsequently, it flattens when training converges to the final model. This suggests that improvement in training can be obtained as long as there is uncertainty in the class labels. Furthermore, tracking the MMS measure may turn out to be useful for designing and monitoring new training regimes.

Our selection criterion was inspired by the Active Learning methods, but our goal, accelerate training, is different. Active learning mainly concerns about the labelling cost. Hence, it is common to keep on training till (almost) convergence, before turning to select additional examples to label. However, such an approach is less efficient when it comes to acceleration. In such a scenario, we can be more aggressive; since labelling cost is not a concern, we can re-select a new batch of examples in each training step.

An efficient implementation is also crucial for gaining speedup. Our scheme provides many opportunities for further acceleration. For example, fine-tuning the sample size used to select and fill up a new batch, to balance between the selection effort conducted at the end of the forward pass, and the compute resources and efforts required to conduct the back-propagation pass. This also opens an opportunity to design and use dedicated hardware for the selection. In the past few years, custom ASIC devices that accelerate the inference phase of neural networks were developed (goy; Hoffer et al.; Jouppi et al., 2017). Furthermore, in (Jacob et al., 2018), it was shown that using quantization for low-precision computation induces little or no degradation in final model accuracy. This observation, together with the fast and efficient inference achieved by ASICs, make them appealing to be used as a supplement accelerator in the forward pass of our selection scheme.

The MMS measure doesn't use the labels. Thus it can be used to select samples in an active learning setting as well. Moreover, similarly to (Jiang et al.) the MMS measure can be implemented at other layers in the deep architecture. This enables to select examples that directly impact training at all levels. The additional compute associated with such calculating and selecting the right batch content, makes it less appealing for acceleration. However, for active learning, it may introduce an additional gain, since the selection criterion chooses examples which are more informative for various layers. The design of a novel Active Learning method is left for further study.

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

## Appendix A

Assume a given point $\mathbf{x}$ and its DNN latest layer output $\mathbf{y} = F(\mathbf{x}, \theta)$. W.l.o.g let's the largest and the second largest scores of the classifier be: $s^1 = \mathbf{w}_1^T \mathbf{y} + b_1$ and $s^2 = \mathbf{w}_2^T \mathbf{y} + b_2$, respectively. We are looking for the smallest $\delta \mathbf{y}$ satisfying:

$$\mathbf{w}_1^T(\mathbf{y} + \delta\mathbf{y}) + b_1 = \mathbf{w}_2^T(\mathbf{y} + \delta\mathbf{y}) + b_2$$

Re-arranging terms we get:

$$-(\mathbf{w}_1 - \mathbf{w}_2)^T \delta\mathbf{y} = (s^1 - s^2)$$

The least-norm solution of the above under-determined equation is calculated using the right pseudo-inverse of $(\mathbf{w}_1 - \mathbf{w}_2)^T$ which gives:

$$\delta\mathbf{y} = -(s^1 - s^2)\frac{\mathbf{w}_1 - \mathbf{w}_2}{\|\mathbf{w}_1 - \mathbf{w}_2\|^2}$$

and therefore the MMS of $\mathbf{y}$ is:

$$d = \|\delta\mathbf{y}\| = \sqrt{\delta\mathbf{y}^T \delta\mathbf{y}} = \frac{s^1 - s^2}{\|\mathbf{w}_1 - \mathbf{w}_2\|}$$

