# OpenReview forum: "Selective sampling for accelerating  training of deep neural networks"
_ICLR.cc/2020/Conference — Reject_

### Official Review · AnonReviewer2 · 2019-10-21
**Official Blind Review #2**

**Rating:** 1

**Review:**

### Summary of contributions

This paper aims to accelerate the training of deep networks using a selective sampling.
They adapt ideas from active learning (which use some form of uncertainty estimation about the class of the label) to selectively choose samples on which to perform the backward pass. Specifically, they use the minimal margin score (MMS).
Their algorithm works by computing the forward pass over a batch of size B (which is much larger than the regular batch of size b), compute the uncertainty measure for each sample, and only perform the backward pass over the b samples with the highest uncertainty. The motivation is that the backward pass is more expensive than the forward pass, and that by only performing this pass on a subset of samples, computations are saved.


### Recommendation

Reject. The central premise of the paper is unclear, the writing/presentation needs improvement, and the experiments are not convincing.


### Detailed comments/improvements:


There is a central premise of the paper that I don't understand: that the forward pass is much cheaper than the backward pass.
This is claimed in the intro by referring to charts that hardware manufacturers publish (but there are no references included), but I don't see why this should be the case.
For a linear network with weights W, the forward pass is given by the matrix-matrix product (rows of X are minibatch samples):
Y = XW^T

and the backward pass is given by the two matrix-matrix products:
dL/dX = dL/dY*dY/dX = dL/dY*W
dL/dW = dL/dY*dY/dW = dL/dY*X^T

Similarly the two operations in the backward pass for convolutional layers are given by a convolution of the output gradients with the transposed weigtht kernels and the input image respectively.

Point being, I don't see why the backward pass should be more than 3x more expensive than the forward pass. A simple experiment in PyTorch confirms this: the code snippet pasted at the bottom shows that the backward pass takes only around 2.6x longer than the forward pass.

fprop: 0.009286s
bprop: 0.0240s
bprop/fprop: 2.5893x

In algorithm 1, it is assumed that b << B. For this to be effective the forward pass would have to be *much* faster than the backward pass for this method to yield an improvement in computation. Can the authors comment on where this justification comes from?

I am unclear on what the purpose of Section 4.1 is. This shows that the MMS of the proposed method is lower than the other two, but this should be completely expected since that is exactly the quantity being minimized.
There are also several unsubstantiated claims: "Lower MMS scores resemble a better...batch of samples", "the batches selected by our method provide a higher value for the training procedure vs. the HNM samples.", "Evidently, the mean MMS provides a clearer perspective...and usefulness of the selected samples". What does higher value, usefulness, clearer perspective mean?

More generally, it is unclear if there is really any improvement in the final performance from using the proposed method.
In Figure 2, all methods seem to have similar final performance.
In Figure 5, is there a reason why the curve for MMS is cut off? How does its final performance compare to that of the baseline method in red? It looks like the baseline might be better, but it's hard to tell from the figure.

Why are the experiments with the entropy measure in a seperate section? Please include them along with the other methods in the same plot, i.e merge Figure 2 and Figure 4.

My suggestions for improving the experimental section are as follows:
- include all methods together in all the plots/tables
- repeat experiments multiple times with different seeds to get error bars. Include these both in the learning curves and in the tables.
- It's hard to see small differences in the learning curves, so including tables as well is important. Include best performance for all the methods in the tables.

Finally, in 2019 CIFAR alone is not longer a sufficient dataset to report experiments on. Please report results on ImageNet as well.

One of the central premises of the paper is acceleration in terms of compute/time. To make this point, there should also be results in terms of walltime and floating-point operations. Please include these results in the paper.




### Code snippet timing forward/backward passes


import torch, torch.nn as nn, time

model =	nn.Sequential(nn.Linear(784, 1000),
                      nn.ReLU(),
                      nn.Linear(1000, 1000),
                      nn.ReLU(),
                      nn.Linear(1000, 10),
                      nn.LogSoftmax())

data = torch.randn(128, 784)
labels = torch.ones(128).long()
t = time.time()
pred = model.forward(data)
loss = nn.functional.nll_loss(pred, labels)
fprop_time = time.time() - t
t = time.time()
loss.backward()
bprop_time = time.time() - t
print('fprop: {:.4}s'.format(fprop_time))
print('bprop: {:.4f}s'.format(bprop_time))
print('bprop/fprop: {:.4f}x'.format(bprop_time / fprop_time))


**Experience Assessment:**

I have published one or two papers in this area.

**Review Assessment: Checking Correctness Of Derivations And Theory:**

N/A

**Review Assessment: Checking Correctness Of Experiments:**

I assessed the sensibility of the experiments.

**Review Assessment: Thoroughness In Paper Reading:**

I read the paper thoroughly.

---

> ### Author Response · Authors · 2019-11-14
> **Training vs. inference and other comments**
>
> Dear reviewer #2:
>
> We would like to thank you for the feedback and the effort involved in running the performance example you presented. We will answer the questions and reply to the comments raised:
>
> 1) As for the answer regarding our central premise. In order to select the samples using our MMS scheme, we leverage inference concepts that are entirely different from training. Some of the prominent ideas are low precision arithmetic operations when applying quantization, layers fusion like Convolution-BatchNorm (applying the BN running statistics into the convolutions and eliminating the need of performing BN) and weight compression. These concepts are not theoretical as they are being used when building specialized hardware accelerators as T4 (by Nvidia), Goya (by Habana) and TPU (by Google), allowing these devices to be ~10X faster at inference than running training step on a modern GPU. A detailed explanation and performance charts can be seen in Google’s TPU paper “In-Datacenter Performance Analysis of a Tensor Processing Unit”.
>
> Additionally, for distributed training on large-scale hardware, the advantage of the inference devices is even greater. As the training instances must wait to a gradient reduction across all instances, the inference devices can perform forward passes on multiple instances in full parallelism (i.e. inference is embarrassingly parallel), without the need to wait to any other instance in the system.  Thus, it can potentially select more offline examples for our MMS scheme.
>
> Finally, more performance benchmarks can be found when referring to Habana’s site (https://habana.ai/inference/ and https://habana.ai/training/) as the training vs. inference throughput on their hardware is 1650 vs. 15453 images/sec.
>
> 2) As for "In Figure 2 all methods seem to have similar final performance.". Our main goal is not to improve final accuracy but rather to train less steps than the vanilla training regime. For that purpose, we introduced the early LR drop regime (as seen in Figure 5). We presented the plots in Figure 2 in order to show the relatively large deviation in the validation error as well as the training error. The validation error decrease using our MMS method implies a faster convergence and that a faster regime can be used. We further accept the comment and will move these plots to the appendix as well as mention their purpose in the paper.
>
> 3) As for cutting CIFAR100 validation error for the early LR drop regime. We decided to end this experiment when the error reaches a sufficient performance w.r.t the baseline training (red). Moreover, we explicitly stated the final accuracy of the baseline and our MMS method in Table 1, showing a drop of 0.07% with almost halving the baseline required steps (from 156K to 80K steps).
>
> 4) We kindly accept your comment regarding the entropy experiment and will include it with the other methods plot.
>
> 5) We couldn't run many experiments due to time limitations, but we will make the effort to add STD bars.
> 6) We will expand Table 1 as suggested with the entropy experiment and will add more relevant step information for clarity.
> 7) We will add the ImageNet experiment.

---

### Official Review · AnonReviewer1 · 2019-10-22
**Official Blind Review #1**

**Rating:** 1

**Review:**

This paper proposes a minimal margin score (MMS) criterion to speed up the training of the deep networks.

I would vote for a clear rejection of this paper. This submission is a clearly unfinished one. The two biggest problems are as follows

1. Lack of a comprehensive discussion on rules for sampling section, please see "Automated Curriculum Learning for Neural Networks". Why previous methods are worse than the proposed one is not clear.

2. All experiments are only compared with baseline approaches. In some experiments, the improvements are really marginal (e.g., Figure 2). In these cases, the STD of these curves is not shown, it is not clear whether the improvements are significant or not.

**Experience Assessment:**

I have published one or two papers in this area.

**Review Assessment: Checking Correctness Of Derivations And Theory:**

I did not assess the derivations or theory.

**Review Assessment: Checking Correctness Of Experiments:**

I assessed the sensibility of the experiments.

**Review Assessment: Thoroughness In Paper Reading:**

I read the paper at least twice and used my best judgement in assessing the paper.

---

### Official Review · AnonReviewer3 · 2019-10-26
**Official Blind Review #3**

**Rating:** 3

**Review:**

A new approach is proposed to speed up training in deep models.

The idea is to select sample batches when back propagating the error based on the distance of the prediction foe the sample from the decision boundary. Specifically, we pick points closer to the boundary, i.e., ones that we are less confident about for backpropagation.

Experiments are performed comparing the method with Hard negative sampling (HNM) , entropy-based sample selection as well as regular training. Experiments are performed on Cifar10 and Cifar100 datasets.
Why only two datasets, the method is general so there should be more datasets to verify its performance.

The results on Cifar100 in Fig 5 c seems to show that we cannot reach the training accuracy using the proposed method as compared to the other methods. What is the intuition here as to why it happens? In general though since the main goal is to speed up training I do not see very convincing evidence of this in the limited evaluation which seems to be the main weakness here.

**Experience Assessment:**

I have read many papers in this area.

**Review Assessment: Checking Correctness Of Derivations And Theory:**

N/A

**Review Assessment: Checking Correctness Of Experiments:**

I assessed the sensibility of the experiments.

**Review Assessment: Thoroughness In Paper Reading:**

I read the paper at least twice and used my best judgement in assessing the paper.

---

> ### Author Response · Authors · 2019-11-14
> **Answer**
>
> Dear reviewer #3:
>
> We would like to thank you for the feedback and will answer the questions raised:
>
> 1) We used CIFAR10 and CIFAR100 to prove our main concept which aims to reduce the number of training steps. We didn't have enough time to include other datasets for the deadline but we plan to add ImageNet to the paper.
>
> 2) Figure 5 and Table 1 show that for CIFAR10 we reached the final accuracy (with a minor drop of 0.25%) after 28% of required training steps using our method. For CIFAR100 we reached the final accuracy (with a minor drop of 0.07%) after 51% of required training steps using our method. This is a very substantial speed up with a very minor drop in final accuracy.

---

### Decision · Program_Chairs · 2019-12-19

**Decision:**

Reject

**Comment:**

The paper proposes a method to speed up training of deep nets by re-weighting samples based on their distance to the decision boundary. However, they paper seems hastily written and the method is not backed by sufficient experimental evidence.